# Does Wim Hof Method Improve Breathing Economy during Exercise?

**DOI:** 10.3390/jcm11082218

**Published:** 2022-04-15

**Authors:** David Marko, Petr Bahenský, Václav Bunc, Gregory J. Grosicki, Joseph D. Vondrasek

**Affiliations:** 1Sports Motor Skills Laboratory, Faculty of Sports, Physical Training and Education, Charles University, 162 52 Prague, Czech Republic; david.marko@seznam.cz (D.M.); bunc@ftvs.cuni.cz (V.B.); 2Department of Sports Studies, Faculty of Education, University of South Bohemia, 371 15 Ceske Budejovice, Czech Republic; 3Biodynamics and Human Performance Center, Georgia Southern University, Savannah, GA 31419, USA; ggrosicki@georgiasouthern.edu (G.J.G.); jv06134@georgiasouthern.edu (J.D.V.)

**Keywords:** Wim Hof method, breathing, runners, adolescents, diaphragm

## Abstract

(1) Background: Breathing economy during endurance sports plays a major role in performance. Poor breathing economy is mainly characterized by excessive breathing frequency (BF) and low tidal volume (V_T_) due to shallow breathing. The purpose of this study was to evaluate whether a 4 week intervention based on the Wim Hof breathing method (WHBM) would improve breathing economy during exercise in adolescent runners. (2) Methods: 19 adolescent (16.6 ± 1.53 years) middle- and long-distance runners (11 boys and 8 girls) participated in the study. Participants were randomly divided into experimental (*n* = 11) and control groups (*n* = 8). The study was set in the transition period between competitive race seasons and both groups had a similar training program in terms of running volume and intensity over the course of the study. The experimental group performed breathing exercises every day (~20 min/day) for 4 weeks. The control group did not perform any kind of breathing exercise. The breathing exercises consisted of three sets of controlled hyperventilation and consecutive maximum breath holds. Before and after the intervention, participants performed incremental cycle ergometer testing sessions consisting of two minute stages at 1, 2, 3, and 4 W·kg^−1^ with breath-by-breath metabolic analysis. During the testing sessions, BF, V_T_, and minute ventilation (V_E_) were assessed and compared. (3) Results: There were no statistically significant differences (*p* > 0.05) in BF, V_T_, or V_E_ between experimental and control groups before or after the intervention. A nonsignificant small-to-large effect for an increase in V_E_ and BF in both groups following the 4 week intervention period was observed, possibly due to a reduction in training volume and intensity owing to the down period between competitive seasons. (4) Conclusions: The 4 week intervention of WHBM did not appear to alter parameters of breathing economy during a maximal graded exercise test in adolescent runners.

## 1. Introduction

There is a dose–response relationship between exercise intensity and oxygen demand [1]. To accommodate this demand, increases in minute ventilation, tidal volume, and breathing frequency are observed as exercise intensity increases. However, increasing ventilation through tidal volume may be most beneficial whereas increasing breathing frequency is associated with a substantial increase in respiratory muscle oxygen cost, and increasing tidal volume does not incur the same oxygen cost and is associated with an increase in oxygen extraction by the alveoli due to slower breathing. A decrease in respiratory rate and increase in tidal volume improves ventilation efficiency via alveolar recruitment and distension, thus reducing alveolar dead space [2,3]. In a trained individual, during submaximal exercise, exhaled air contains only 14–15% oxygen, whereas exhaled air in an untrained person under the same load intensity contains up to 17% oxygen. Breathing economy can be improved by exercise [4]. For this reason, an untrained person must breathe more air to achieve the same submaximal oxygen absorption [5]. Because of shallow breathing, there is a higher demand for the energy needed to ventilate, but there is also a greater amount of dead space ventilation (DSV), a volume of air that does not engage in gaseous exchange with the blood [5].

Focused breath training can improve strength and endurance respiratory muscles, thereby improving breathing phenotype [5,6,7,8,9]. Respiratory muscles use up to 11% of total energy output during maximal exercise [5]. However, by increasing tidal volume and reducing breathing frequency, performance can be improved through an attenuation of respiratory muscle fatigue, which leaves more energy for non-respiratory muscles [5,10].

The Wim Hof method was founded by the Dutchman Wim Hof, holder of almost 20 world records. This method stands on three pillars—breathing exercises, cold exposure, and meditation [11]. Breathing exercises consist of deep breaths and subsequent breath holds that are performed after exhaling. Each breath is performed using a yoga breath wave that starts in the abdomen (diaphragmatic breathing) and continues into the chest [11,12]. Reducing breathing volume and holding the breath after exhaling reduces lung hyperinflation, which leads to lower residual lung volume [13]. Dynamic hyperinflation may be the reason for the disruption of the neuromechanical aspects of breath control and diaphragm dysfunction [13]. Other benefits of repeated breath restraints are improved respiratory motor plasticity, improved upper airway tone by increasing the activity of the vagus nerve and the hypoglossal nerve, improved oxygen metabolism, reduced inflammation, and/or increased stress tolerance [13].

The aim of this study was to investigate the effect of one of the pillars of the Wim Hof method—breathing exercise—on breathing economy during dynamic work. We hypothesized that breathing exercises based off of the Wim Hof method would significantly change the breathing pattern during exercise.

## 2. Materials and Methods

### 2.1. Participants

The study involved 19 adolescent middle- and long-distance runners, 11 of which were boys and 8 were girls. The average age was 16.6 ± 1.53 years, the average weight was 62.5 ± 9.34 kg, and the average height was 174.58 ± 8.60 cm. All of the recruited runners competed at the elite youth national level for at least three years. Participants were taken from one run training group and all athletes had comparable fitness levels. Participants were randomized into an experimental and control group using a research randomizer [14]. The experimental group consisted of 11 participants (6 boys and 5 girls) and the control group consisted of 8 participants (4 boys and 4 girls). Inclusion criteria were: age of 14–19 years, performance level (attendance in national championship), and good health, defined as free from known cardiovascular, metabolic, and/or renal disease. Excluding criteria were: asthma, hypertension, or any known disease. During the study, participants were in a period between competitive seasons. The running training intensity was lower than in a competitive period (lower by 20%), as was the volume of running training. However, total training volume was similar to usual as well as the total training intensity. Though runs were still performed, athletes substituted runs with other activities such as swimming, cycling, hiking, games, etc. The non-specific load with a relatively small external load can cause a large submaximum internal load of the athlete. The training program was the same for the experimental and control group in terms of intensity and volume.

### 2.2. Protocol

Participants were asked to complete baseline testing, 4 week interventional program, and post-intervention testing. The test and re-test consisted of a graded exercise test (GXT) on a bicycle ergometer (Lode Excalibur 38k4, Lode B.V., Groningen, The Netherlands) with subsequent breath-by-breath metabolic analysis (Metalyzer B3, Cortex, Leipzig, Germany). After an initial warmup at 25 W and a cadence of 80 rev∙min^−1^, participants completed four, two minute stages at 1, 2, 3, and 4 W∙kg^−1^, as previously described [8,9,15]. Participants were instructed to maintain a cadence of ~100 rev∙min^−1^ throughout the test. After the last workload phase (4 W∙kg^−1^), a three minute cooldown with a cadence of 60 rev∙min^−1^ and a resistance of 25 W followed. During the GXT, the following outcomes were measured: tidal volume (V_T_), minute ventilation (V_E_), and breathing frequency (BF).

### 2.3. Breathing Exercise

Breathing exercises were performed in a supine position. This position was recommended by Hof himself to ensure maximal safety [16]. Breathing exercise consisted of 30 full breaths at a metronome rhythm of 20 breaths·min^−1^; subsequently, participants fully exhaled and held their breath until they felt an urgent need to breathe or until the first spontaneous contraction of the diaphragm. Every breath hold was immediately followed by another 15 s breath hold preceded by one full breath. The cycle was then repeated twice more without a break. Overall, the breathing exercise lasted 17–22 min depending on individual breath holding duration. Participants were instructed to exercise on daily basis and with an empty stomach. The control group was instructed to stay in a supine position for ~20 min and breath naturally.

### 2.4. Statistical Analysis

The normality of data was confirmed using the Shapiro–Wilk test. Data are presented as mean ± standard deviation. A two-way repeated-measures ANOVA (group × time) was used to compare outcome measures. Significant interactions were examined using Bonferroni-adjusted simple main effect post-hoc comparisons. An alpha-level of 0.05 was used to assess statistical significance for all comparisons. Subsequently, effect size was determined using Cohen’s d. The data processing was done in Excel 2016 (Microsoft, Oregon, WA, USA) and Statistica 12 (StatSoft, Tulsa, OK, USA).

## 3. Results

Ventilation parameters in the experimental and control groups are displayed in Table 1 and Table 2, respectively. The minute ventilation parameter showed no statistical significance between experimental and control group *p* = 0.138; *p* = 0.825; *p* = 0.479; *p* = 0.489 in each load stage, i.e., 1, 2, 3, or 4 W·kg^−1^. A small-to-large effect for an increase in minute ventilation was observed post-intervention in both groups. As for the tidal volume parameter, no significance was found in all load stages in between the experimental and control group (*p* = 0.630; *p* = 0.377; *p* = 0.688; *p* = 0.087). Regarding breathing frequency, once again, no statistically significant difference was observed between groups (*p* = 0.794; *p* = 0.917; *p* = 0.956; *p* = 0.296). Small-to-moderate effects for changes in breathing frequency were noted in both groups post-intervention.

## 4. Discussion

The primary aim of this study was to investigate the effect of a 4 week application of the Wim Hof breathing method (WHBM) on ventilation parameters during graded exercise testing in adolescent athletes. To date, a number of studies have demonstrated performance benefits associated with Wim Hof Breath training [17,18,19,20]. For instance, our group recently reported accelerated oxygen consumption kinetics and reductions in heart rate during submaximal cycling exercise following a Wim Hof-based breathing intervention [20]. Another study investigated the acute effect of a single Wim Hof Breathing session on anaerobic performance using repeated sprints, though no benefits were observed [21]. To the best of our knowledge, this study is the first to examine whether Him Hof breath training may influence ventilatory parameters (V_T_, V_E_, and BF) during graded exercise testing.

Respiratory muscle training can improve endurance and strength of respiratory muscles [5,22]. Studies that have tried to use respiratory muscle training as a possible tool for enhancing performance when performing whole-body exercises most often used one of the following training options: voluntary isocapnic hyperpnea, flow resistive loading, and pressure threshold loading. All of the mentioned training modes have the potential to improve specific aspects of respiratory muscle function [23]. However, a critical look at previous studies reveals that the effect of respiratory muscle training on exercise performance is not uniform [22,23,24]. This study did not measure endurance and strength of respiratory muscles directly, but it seems that because there is no significant improvement in ventilation parameters, there was no improvement in breathing muscles due to Wim Hof breath training. As these were young and fit individuals, it is possible there were no noticeable increases in respiratory muscle strength due to the “ceiling effect”.

During exercise, there is an increase in minute ventilation due to an increase in tidal volume and breathing frequency [22]. More economical, however, is an increase in primarily tidal volume, not in breathing rate, which leads to an acceleration of respiratory muscle fatigue. The consequences of overworking respiratory muscles and the diaphragm are vasoconstriction and blood flow reduction affecting working locomotor muscles [25]. Another putative mechanism of improving performance due to respiratory muscle training is a decrease in the perception of respiratory and limb discomfort or redistribution of blood flow from respiratory to locomotor muscles [23]. There are a number of studies trying to influence and streamline ventilation in endurance athletes [26,27,28,29]. In the study of Sonetti et al. [28], competitive male cyclists completed a 5 week respiratory muscle training program with 30–35 min/day sessions 5 days/week. Training session consisted of 50 to 60% of 15 s maximal voluntary ventilation (MVV) using a breathing frequency of 50 to 60 breaths/min for 30 min. This protocol was not efficient in improving 15 s MVV or endurance breathing performance at 90% of MVV_15_. This study using the Wim Hof breathing technique was also not efficient on elite adolescent runners. It is possible that the effect of interventional breathing exercises may not be as effective in elite athletes. Further research can investigate these methods in sedentary individuals or clinical populations where greater benefits may be realized. Limitations of this study are a lack of prior sample size calculations and the use of a bicycle ergometer as it does not fully demonstrate specific terms of running.

## 5. Conclusions

The 4 week intervention exercise based on the pillar of the Wim Hof breathing method was not effective in improving the breathing economy of adolescent elite endurance runners. The breathing program did not significantly alter tidal volume or reduce breathing frequency and thus did not save the energy required for breathing muscle work. Further research may focus on more extended breath intervention or choose untrained populations. To this end, it may be more appropriate to use already proven intervention programs to improve breathing economy, such as yoga-based breathing exercises [7,8,9] or inspiratory muscle training [30,31].

## Figures and Tables

**Table 1 jcm-11-02218-t001:** Ventilation parameters before and after intervention in experimental group (*n* = 11).

	Workload	Before Intervention	After Intervention	% Change	Cohen’s *d*
V_E_ [L·min^−1^]	1 W·kg^−1^	38.08 ± 4.94	42.91 ± 7.27	12.57 ± 11.65	0.77
	2 W·kg^−1^	52.26 ± 5.59	57.23 ± 6.49	10.47 ± 15.76	0.81
	3 W·kg^−1^	72.81 ± 6.83	80.66 ± 10.21	11.15 ± 13.22	0.90
	4 W·kg^−1^	96.87 ± 13.42	101.93 ± 14.74	5.91 ± 14.45	0.35
	**MEANS**	**65.01 ± 22.16**	**70.68 ± 22.52**	**10.02 ± 2.49**	
V_T_ [L]	1 W·kg^−1^	1.46 ± 0.50	1.43 ± 0.34	0.47 ± 11.76	−0.08
	2 W·kg^−1^	1.78 ± 0.51	1.76 ± 0.43	0.05 ± 8.42	−0.04
	3 W·kg^−1^	2.05 ± 0.53	2.08 ± 0.45	2.67 ± 8.58	0.05
	4 W·kg^−1^	2.25 ± 0.57	2.32 ± 0.63	3.08 ± 7.05	0.12
	**MEANS**	**1.89 ± 0.30**	**1.90 ± 0.34**	**1.57 ± 1.72**	
BF [breath·min^−1^]	1 W·kg^−1^	28.11 ± 7.44	31.30 ± 7.44	13.46 ± 16.48	0.42
	2 W·kg^−1^	31.41 ± 8.66	34.01 ± 8.15	11.04 ± 17.08	0.30
	3 W·kg^−1^	37.55 ± 8.84	39.93 ± 7.21	9.07 ± 17.38	0.29
	4 W·kg^−1^	44.69 ± 8.75	45.99 ± 9.83	2.75 ± 11.11	0.13
	**MEANS**	**35.44 ± 6.32**	**37.81 ± 5.66**	**9.08 ± 3.97**	

V_T_: tidal volume; V_E_: minute ventilation; BF: breathing frequency.

**Table 2 jcm-11-02218-t002:** Ventilation parameters before and after intervention in control group (*n* = 8).

	Workload	Before Intervention	After Intervention	% Change	Cohen’s *d*
V_E_ [L·min^−1^]	1 W·kg^−1^	39.22 ± 4.15	40.98 ± 6.57	4.06 ± 8.93	0.31
	2 W·kg^−1^	52.90 ± 7.50	58.64 ± 11.40	10.36 ± 11.51	0.59
	3 W·kg^−1^	76.67 ± 10.03	81.52 ± 15.13	5.93 ± 9.63	0.37
	4 W·kg^−1^	94.86 ± 9.08	96.36 ± 11.13	1.51 ± 5.41	0.14
	**MEANS**	**65.91 ± 21.42**	**69.38 ± 21.20**	**5.46 ± 3.23**	
V_T_ [L]	1 W·kg^−1^	1.27 ± 0.13	1.28 ± 0.24	−0.02 ± 9.22	0.04
	2 W·kg^−1^	1.57 ± 0.26	1.61 ± 0.33	2.48 ± 9.86	0.14
	3 W·kg^−1^	1.85 ± 0.23	1.85 ± 0.30	−0.58 ± 7.08	−0.01
	4 W·kg^−1^	1.96 ± 0.20	1.92 ± 0.23	−2.49 ± 5.91	−0.21
	**MEANS**	**1.66 ± 0.27**	**1.67 ± 0.25**	**−0.14 ± 1.76**	
BF [breath·min^−1^]	1 W·kg^−1^	31.00 ± 4.57	32.51 ± 6.25	4.48 ± 8.69	0.27
	2 W·kg^−1^	34.09 ± 4.44	36.82 ± 5.83	8.28 ± 12.42	0.52
	3 W·kg^−1^	41.57 ± 5.39	44.23 ± 5.84	7.02 ± 11.81	0.47
	4 W·kg^−1^	48.38 ± 3.97	50.29 ± 3.08	4.38 ± 8.01	0.53
	**MEANS**	**38.76 ± 6.75**	**40.96 ± 6.82**	**6.04 ± 1.66**	

V_T_: tidal volume; V_E_: minute ventilation; BF: breathing frequency.

## Data Availability

Data sharing is not applicable.

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
