# Peer review of "Does Wim Hof Method Improve Breathing Economy during Exercise?"

_jcm, 2022, doi:10.3390/jcm11082218_

Round 1
Reviewer 1 Report
The authors investigated the efficacy of the Wim Hof Method breathing (WHBM) in enhancing the breathing economy of adolescent runners during exercises. Participants were distributed into experimental and control groups in which the experiment group received a 4-week WHBM intervention along with a normal training routine. No statistically significant difference was reported in the breathing pattern between the control and experimental groups. It was postulated that the 4-week intervention of WHBM did not enforce changes in the breathing parameters of the participants. Overall, the manuscript is an intreating read. However, a few issues need to be clarified.
L 35-46 any evidence to support these statements?
No information was reported on the determination of the appropriate sample size for the study. How does the sample size determine?
Including and excluding criteria of the participants should be clearly explained
Was the fitness level of the participants measured and found to be equal before the commencement of the study?
The authors should detail the process of allotting the participants into groups
A new section should be developed to highlight the practical implications of the study findings as well as possible limitations.
Reviewer 2 Report
The manuscript submitted for review touches on the interesting issue of teaching breathing technique. The context of breathing economy is crucial in sport. This is especially true in those disciplines where intensification of effort leads to loss of breathing rationalization. The design of the experiment is correct. The organization of the experiment is appropriate. However, two questions should be asked at this point:
Why were endurance athletes, who have a well mastered breathing technique, chosen for the experiment? It is unlikely that the experimental procedure would work in such a group.
Why was a cycloergometer exercise, which is highly unspecific for runners, used? Was this an impediment to correct (training-trained) breathing?
These questions need clarification.
The location of the experiment during the transition period is inherently correct. However, the authors forgot (did not take into account) that the work of respiratory muscles accompanies every training as well as the musculoskeletal system. Therefore, a decrease in the strength of the training stimulus must contribute to a decrease in the efficiency of the respiratory muscles. Therefore, could progressive changes in breathing economy be expected in such a case? In my opinion, there is no support in the paper for such a hypothesis. Was the experimental program supposed to inhibit the decline in respiratory muscle efficiency? Also no justification for such a hypothesis.
The paper is well structured, but it is hard to resist the impression that it was not prepared in terms of posing a solid research problem. The authors' previous research itself stands in opposition to the current results. But the question remains. Did the choice of the group and the time of conducting the experiment allow to reveal the expected changes. In my opinion it did not and here a mistake was made. I ask the authors to convince me that my opinion can be challenged in light of the literature of the study. We are talking about athletes of endurance sports evaluated in non-specific effort in terms of breathing economy during a period of strongly reduced training intensity for 4 weeks.
Round 2
Reviewer 1 Report
The authors have sufficiently addressed the concerns raised during the initial review
Reviewer 2 Report
Thank you very much for clarifying and completing the text of the paper.
The essence of the paper requires an unambiguous definition of training loads during the transition period. The authors refer to volume. The problem of breathing and its economy appears with the increase of intensity. Please complete the paper by indicating to what extent the intensity was lowered during the transition period. It is very important because of the obtained effect of the experiment. I would like to suggest to the authors that the use of non-specific load can with a relatively small external load cause a large-submaximum internal load of the athlete.
